# Numerical and Experimental Investigations of Micro Thermal Performance in a Tube with Delta Winglet Pairs

**DOI:** 10.3390/mi12070786

**Published:** 2021-06-30

**Authors:** Jiangbo Wang, Ting Fu, Liangcai Zeng, Guang Chen, Fue-sang Lien

**Affiliations:** 1Key Laboratory of Metallurgical Equipment and Control Technology, Wuhan University of Science and Technology, Wuhan 430081, China; qdgwbs@163.com (J.W.); zengliangcai@wust.edu.cn (L.Z.); 2Ministry of Education & Hubei Key Laboratory of Mechanical Transmission and Manufacturing Engineering, Wuhan University of Science and Technology, Wuhan 430081, China; 3Key Laboratory of Traffic Safety on Track, Central South University, Ministry of Education, Changsha 410000, China; gszxcgsp@163.com; 4Department of Mechanical & Mechatronics Engineering, University of Waterloo, Waterloo, ON N2L 3G1, Canada; fslien@uwaterloo.cn

**Keywords:** new vortex generators, heat transfer enhancement, multi-longitudinal vortices

## Abstract

In this research, a novel vortex generator (VG) is presented. The experimental and numerical investigations were carried out to study the micro thermal-hydraulic performance in a heated tube. The numerical results showed that the fluid in the core flow region and the near-wall region was fully mixed because of the longitudinal vortices created by the vortex generators. In addition, the experimental results showed that the heat transfer coefficient (h) decreased with the increasing pitch ratio (PR) value, while the friction coefficient exhibited the opposite trend. With the increasing ration angle (RA) numbers, the h values decreased while the f numbers increased. In addition, the maximum and minimum values of the fraction ratio were 1.66 and 4.27, while these values of the Nusselt number ratio were 1.24 and 1.83. The maximum thermal enhancement factor (TEF) was 1.21 when PR = 0.5, RA = 0° and Re = 9090. The heat transfer enhancement mechanism of the vortex generator is explained from the microscopic point of view.

## 1. Introduction

With the increase in serious energy problems in recent years, higher requirements for energy utilization and recovery have been put forward. The convective heat transfer method was widely used to improve energy efficiency by enhancing heat transfer efficiency in the process of energy consumption. 

Heat transfer enhancement technology of single-phase flow was generally divided into three categories [1] and the passive method was more common in practical applications for low cost, easy maintenance and no reliance on external energy contrast of the active method and compound method [2,3]. Vortex generators (VGs) which were placed in tubes of heat exchangers, were regarded as an important passive method to enrich energy efficiency. The role of the VGs was to create multi-longitudinal vortices (MLVs), which could result in better mixing of the fluid and restrain the growth of the thermal boundary layer. Thus, the MLVs played a decisive role in enhancing heat exchange efficiency [4]. 

In fact, wings [5] and winglets [6] were widely used due to many advantages, including simple to manufacture, low cost, and low-pressure drop [7,8,9]. Ke et al. [10] numerically investigated the heat transfer performance (HTP) of different winglet arrangements in a rectangular channel. They believed that the channel height and the aspect ratio of VGs were the two major parts affecting the TEF. They also used the “Method of Images” to analyze the dynamic characteristics of the MLVs in the channel. Sun et al. [11] studied the influence of three arguments of RWVGs on the HTP in a heated tube. They reported that the height of RWVG s had a great impact on the induced MLVs, and the HTP was enriched significantly with a lower pitch ratio of the RWVG’s. The influence of different delta winglet arrangements on HTP was studied by Khoshvaght-Aliabadi et al. [12]. Their experimental results showed that the heat transfer coefficient (*h*) and friction factor (*f*) increased with the application of VGs. The best TEF was 1.41 when the Reynold number (*Re*) was 8715. 

Xu et al. [13] discussed the internal flow behaviors and different distributions of Nu, f and TKE caused by VGs with different attack angles and blocking ratios numerically. The effects of different structures of VGs on the improvement of overall thermal performance were explained based on the estimation of the axial turbulent wake. Liang et al. [14] found that the WVG installed in the tube could produce MLVs and provided a higher level of turbulence with relatively low-pressure loss. Subsequently, Xu [15] and Zhai [16] conducted experimental studies on the thermal behavior of winglets. To better explore the basic mechanism of heat transfer, they demonstrated MLVs through a high-speed camera and a smoke generator. Pourhedayat et al. [17] carried out a numerical study on the new positioning of the DWVGs applied in a tube. The HTP of DWVGs with backward and forward configurations were analyzed. Hatami et al. [18] proposed a new heat exchanger for diesel engines. 20 VGs were applied with the best size and angle of attack in the exhaust. The Nu, f and exergy recovery of the simple two-tube heat exchanger were compared with those of the simple two-tube heat exchanger. Oneissi et al. [19] numerical studied the thermal behavior of IPWPVGs. They pointed that the IPWP has lower pressure loss because of its special design.

To improve the HTP of the VGs, some new VGs were proposed and studied. He et al. [20] numerically simulated the HTP of plate-fin tube-shell heat exchangers. On the basis of the field synergy principle (FSP), they thought that reducing the crossing angle between two fields including the velocity field and temperature gradient field was the underlying mechanism of strengthening convective heat exchange. They also suggested that slotting and other techniques should be used at the rear of the fin to improve the coordination of the two fields. Baissi et al. [21] investigated the HTP and energy loss of a SAH channel fitted with VGs experimentally. After analyzing the flow structure of perforated and non-perforated curved delta-shaped VGs, they found that the perforated VGs suffered relative lower pressure loss penalty, but the non-perforated delta-shaped VGs had a maximum TEF of 2.21. Lu et al. [22] studied the flow characteristics of punched planar VGs and curved VGs, respectively. They believed that punching holes on the surface of VGs could achieve better HTP, and the better hydraulic performance was achieved at a lower position and near the leading edge of the punched holes. Skullong et al. [23] carried out an experimental study on the convective heat transfer in a heated tube by placing several pairs of perforated delta winglets repeatedly on the perforated-cross tape. They found that the f values were decreased with the punched holes.

Song et al. [24] studied the HTP of the concave and convex curved VG under the condition of laminar flow in a rectangular channel at different central angles ($10^{{}^{\circ}}-80^{{}^{\circ}}$) and three different attack angles ($20^{{}^{\circ}},30^{{}^{\circ}},40^{{}^{\circ}})$. The numerical result showed that the HTP of the concave curved VG was best when the central angle was greater than 60° and the angle of attack was 20°, while the HTP of convex curved VG was inhibited. Agung [25] experimentally studied the HTP of the tube with the insertion of punched delta-winglets (PDWVGs) at 5500<Re<14,500. Three different attack angles ($\alpha =30^{{}^{\circ}},50^{{}^{\circ}},\;70^{{}^{\circ}}$), pitch ratio (1.05), and width ratio (0.42) of the PDWVGs were investigated, respectively. The results showed that the f values increased as α increased. The maximum TEF was found to be 1.22 when α was 70°. Another punched VGs with holes was studied by Gupta [26], they pointed that punched winglets with different positions had different effects on the overall HTP.

Thus, from the above literature we could find that the MLVs induced by the VGs could improve the HTP significantly. Under the excitation of the insert of WVG, a new type of VG was proposed, which was composed of two delta winglets and a backward inclination angle (namely, DWPVG), and the structure of MLVs created by the DWPVGs and their effect on the HTP was studied. Besides, the influence of three pitch ratios (PR=l/D=0.5,1.0,1.5) and the four rotation angles ($RA=\beta =0^{{}^{\circ}},3.3^{{}^{\circ}},6.7^{{}^{\circ}},10^{{}^{\circ}}$) of the DWPVGs on the HTP were investigated. The results were explained based on the convective transport equation of heat flux from the micro point of view, respectively. The TEF of this work was then compared with the work of predecessors.

## 2. Model Description

### 2.1. Details of the DWPVGs

The detail of the designed DWPVGs was depicted in Figure 1. ABS material was selected to form the DWPVGs by 3D print method, which was shown in Figure 1a. Each DWPVG was composed of two identical triangular winglets with a backward inclination angle. The dimensions of a single DWPVG were shown in Figure 1b. These DWPVGs were mounted on rings (5 mm width and 1.5 mm height) with two thin rods (2 mm diameter) were applied to connect these rings. Figure 1c shows the diagram of the test section with DWPVGs which were connected with rings and rods with glue (HY-106AB). The schematic diagram of DWPVGs placed in the tube is shown in Figure 1d. The DWPVGs were 150 mm away from the entrance of the test section. Every six DWPVGs were uniformly arranged in the circumferential direction of the tube. Two planes were located at x1 = 1735 mm and x2 = 1740 mm for further discussion. Along the x-direction, the rotation angle of the down circumferential arranged DWPVGs relative to the up DWPVGs was defined as  β. The heat transfer characteristics were discussed with three pitch ratios (PR=l/D=0.5,1.0,1.5) and four rotation angles ($RA=\beta =0^{{}^{\circ}},3.3^{{}^{\circ}},6.7^{{}^{\circ}},10^{{}^{\circ}})$, respectively.

### 2.2. Experimental Setup

The experiments were performed in this section and the schematic of the experimental facility was presented in Figure 2. The experimental facility compromised an air blower and three sections (namely upstream section, test section, and downstream section). The tube was made of 304 stainless steel with an inner diameter of 51 mm and 1.5 mm thickness. The lengths of the three sections were 1500 mm, 500 mm and 200 mm, respectively. The length of the upstream section was set to 1500 mm to reduce the uneven flow in the experiment. The fluid was in a fully developed state when it entered the test section. The parts were connected by a PC flange. The speed of the blower was adjusted through the voltage stabilizer and the voltage regulating transformer. The air flow into the tube was controlled through the valve, and the wind speed at the exit of the tube was measure using an electronic anemometer (hf8120). Two 3 mm micro holes were turned both in the inlet and outlet of the test section and connected with the pressure gauge through the hose. The pressure drop between the two sides was measured through the pressure converter. The flexible hose was winded at the outside of the tube. Hot water was circulated through pumps and hoses and insulation was used to keep the pipe at a constant high temperature condition of the tube. Eight T-type thermocouples were uniformly attached to the inner wall of the tube through a small hole with a diameter of 2 mm. Two T-type thermocouples were placed at the entrance and outlet of the test section to measure the temperature, respectively. Foam copper was also used at the exit of the test section to improve the measurement accuracy. The experimental device was placed in a closed room. The environment temperature was adjusted through air conditioning to make sure the temperature at the entrance of the tube was uniform and equal to the room temperature. All temperature data were connected to a multi-channel data recorder and displayed on the PC. 

### 2.3. Data Reduction and Uncertainty Analysis

The heat transfer rate in the air-side was determined as follows [27]:(1)$$Q=\tilde{m}\times C_{p}\times \left(T_{m,out}-T_{m,in}\right),$$
where, $\tilde{m}$ and $C_{p}$ represented the mass flow rate and specific heat of the air at the bulk temperature, respectively. $T_{m,out}$ and $T_{m,in}$ were the average temperature of the cross-section located at the inlet and outlet of the test section, and calculated as follows:(2)$$T_{m,out}=\left(T_{m,out,1}+T_{m,out,2}\right)/2;$$
(3)$$T_{m,in}=\left(T_{m,in,1}+T_{m,in,2}\right)/2.$$

The heat transfer coefficient was determined by:(4)$$h_{t}=Q/A\left(T_{inner,wall}-T_{b}\right).$$

In Equation (4), A and $T_{inner,wall}$ are the area and mean temperature of the inner surface of the test section, $T_{b}$ is the bulk temperature of the test section and the $T_{inner,wall}$ and $T_{b}$ are given by:(5)$$T_{inner,wall}=\sum^{\;}T_{inner,wall}/8;$$
(6)$$T_{b}=\left(T_{in}+T_{out}\right)/2.$$

The Re and f are defined as:(7)$$Re=\rho u_{in}d_{c}/\mu ;$$
(8)$$f=2\times \Delta p\times d_{c}/\left(L\rho u_{in}{}^{2}\right).$$

In the above equations, ρ and μ are the density and thermal conductivity of air based on $T_{b}$. $u_{in}$ is the mean velocity measured at the entrance of the upstream section. $d_{c}$ and L denote the hydraulic diameter of the tube and the length of the section, respectively. Δp represents the pressure drop between the inlet and outlet of the test section.

The *Nu* is determined by:(9)$$Nu=h\times \frac{d_{c}}{\lambda },$$
where λ is the thermal conductivity of the air based on the $T_{b}$.

The TEF [28,29] is given by: (10)$$TEF=\left(\frac{Nu}{Nu_{0}}\right){\left(\frac{f}{f_{0}}\right)}^{-\frac{1}{3}},$$
where $Nu_{0}$ and $f_{0}$ are the Nusselt and friction of tube absence of DWPVGs, respectively.

The method recommended by the literature [30,31] was used to analyze the uncertainties of the present experiment. The maximum uncertainties of three parameters including Re, f and Nu were 0.2%, 2.95% and 7.07%, respectively. 

## 3. Numerical Simulation

### 3.1. Simulation Method

The fluid flow and the HTP of the heated tube with a plurality of DWPVGs were conducted to clearly understand the mechanism of the heat transfer enhancement. To simplify the simulation, we made the following assumptions: The medium (air) was Newtonian and had constant properties.Viscous dissipation and gravity were ignored.The insert was considered rigid.

In the turbulence simulation, the appropriate turbulence model played a decisive role in the simulation results. The widely used models including Realizable k−ε, Standard k−ε, Standard k−ω and SST k−ω are compared in Figure 3. The simulation results of SST k−ω model proposed by [32] are in good agreement with the experimental results with a maximum error of 3.99%. Thus, the SST k−ω model was selected for further investigation. The governing equations are available in the literature [33].

All governing equations were solved by commercial CFD software STAR CCM+, which is based on the finite volume method. The SIMPLE algorithm was selected to deal with the pressure velocity coupling problem. The energy, momentum and turbulence equations were discretized by the second-order upwind scheme, and the solutions were considered convergent when the residuals of the continuity equation, momentum equation and turbulence equation were less than $10^{-5}$ and the energy equation was less than $10^{-7}$.

### 3.2. Boundary Conditions and Numerical Method

In the numerical simulation, the Re number ranged from 9090 to 16,660. The DWPVGs were set to be insulated and the boundary conditions adopted in the three sections are listed in Table 1.

STAR CCM+ commercial software was used to generate the unstructured grid (see Figure 4). To obtain accurate results within the boundary layer of all surfaces, grid refinement was implemented, and the largest value of y+ was less than 1.

As illustrated in Figure 5, four different grid sizes were checked off: $PR=1.5,\;\;RA=10^{{}^{\circ}}$ with 3 million, 7 million, 11 million and 19 million at Re=9090, respectively. The number of grids increased from 11,220,383 to 19,119,226 and the maximum deviations of Nu and f were 0.012% and 0.16%, respectively. Therefore, the grid system with 11,220,383 was selected for subsequent simulations.

### 3.3. Verification of the Smooth Tube

Nu and f of the experimental results of the smooth tube were served to verify the precision of the results with the following well-established empirical correlations:(11)$$Nu=\frac{\left(f/8\right)\left(Re-1000\right)Pr}{1+12.7{\left(f/8\right)}^{1/2}\left({Pr}^{2/3}-1\right)}\times \left(1+{\left(\frac{D}{L}\right)}^{2/3}\right)$$

Gnielinski correlation [34];
(12)$$Nu=Nu_{m,T}\left(Re=2300\right)+\frac{\left(f/8\right)\left(Re-2300\right)Pr^{1.008}}{1.08+12.39{\left(f/8\right)}^{1/2}\left({Pr}^{2/3}-1\right)}\times \left(1+{\left(\frac{D}{L}\right)}^{2/3}\right)$$

Taler correlation [35];
(13)$$f={\left(0.79lnRe-1.64\right)}^{-2}$$

Petukhov correlation [36];
(14)$$f=0.316Re^{-0.25}$$

Blasius correlation [37];
(15)$$f={\left(1.2776logRe-0.406\right)}^{-0.25}$$

Taler correlation [38].

As illustrated in Figure 6, the Nu and f of the experimental system established were in accordance well with the empirical correlations. Compared to Gnielinski and Talor correlations, the maximum deviation of Nu were 16.2% and 17.8%, while the maximum error of f related to Petukhov, Blasius and Taler correlations were 7.71%, 7.89% and 8.06%, respectively, which could mean the model established in this paper was valid. 

## 4. Results and Discussion

### 4.1. Effects of VG’s Arrangement Styles on Heat Transfer Coefficient

Figure 7a shows the relationship of Re and h for these tubes with different PRs, RAs and Res. The experimental results indicate that the values of h were higher for the tubes with VGs compared to smooth tubes due to the existence of MLVs from VGs. Figure 7b shows the relationship of RA and h when the values of PR and Re are the same. The curves explained that the values of h decreased with an increase in the values of RA. This means the heat transfer performance was restrained when the values of RA increased. Figure 7c shows the relationship of PR and h when the values of RA and Re are the same, highlighting that the HTP was poor when the PR values increased.

The h was determined by heat flux q, and heat flux q was determined by velocity, velocity gradient and the combination of velocity and velocity gradient in liquid convective transport. More information can be found in [39].

The convective transport equation of heat flux for steady state flow was as follows:(16)$$\left(v\cdot ∆\right)q+e_{ij}\cdot q=\alpha \nabla^{2}q;$$
where q=−λ∇T and $e_{ij}=\nabla v$, λ represented thermal conductivity.

In the three directions (x, y, z), the Equation (16) could be written as:(17)$$\left(u\frac{\partial q_{x}}{\partial x}+v\frac{\partial q_{x}}{\partial y}+w\frac{\partial q_{x}}{\partial z}\right)+\left(q_{x}\frac{\partial u}{\partial x}+q_{y}\frac{\partial v}{\partial x}+q_{z}\frac{\partial w}{\partial x}\right)=\alpha \nabla^{2}q_{x};$$
(18)$$\left(u\frac{\partial q_{y}}{\partial x}+v\frac{\partial q_{y}}{\partial y}+w\frac{\partial q_{y}}{\partial z}\right)+\left(q_{x}\frac{\partial u}{\partial y}+q_{y}\frac{\partial v}{\partial y}+q_{z}\frac{\partial w}{\partial y}\right)=\alpha \nabla^{2}q_{y};$$
(19)$$\left(u\frac{\partial q_{z}}{\partial x}+v\frac{\partial q_{z}}{\partial y}+w\frac{\partial q_{z}}{\partial z}\right)+\left(q_{x}\frac{\partial u}{\partial z}+q_{y}\frac{\partial v}{\partial z}+q_{z}\frac{\partial w}{\partial z}\right)=\alpha \nabla^{2}q_{z}.$$

Equations (17)–(19) could be rewritten further as follows:(20)$$w_{c-x}+w_{e-x}=\alpha \nabla^{2}q_{x};$$
(21)$$w_{c-y}+w_{e-y}=\alpha \nabla^{2}q_{y};$$
(22)$$w_{c-z}+w_{e-z}=\alpha \nabla^{2}q_{z};$$
where $w_{cx}$, $w_{ex}$, $w_{cy}$, $w_{ey}$, $w_{cz}$ and $w_{ez}$ are defined by:(23)$$w_{c-x}=u\frac{\partial q_{x}}{\partial x}+v\frac{\partial q_{x}}{\partial y}+w\frac{\partial q_{x}}{\partial z};$$
(24)$$w_{e-x}=q_{x}\frac{\partial u}{\partial x}+q_{y}\frac{\partial v}{\partial x}+q_{z}\frac{\partial w}{\partial x};$$
(25)$$w_{c-y}=u\frac{\partial q_{y}}{\partial x}+v\frac{\partial q_{y}}{\partial y}+w\frac{\partial q_{y}}{\partial z};$$
(26)$$w_{e-y}=q_{x}\frac{\partial u}{\partial y}+q_{y}\frac{\partial v}{\partial y}+q_{z}\frac{\partial w}{\partial y};$$
(27)$$w_{c-z}=u\frac{\partial q_{z}}{\partial x}+v\frac{\partial q_{z}}{\partial y}+w\frac{\partial q_{z}}{\partial z};$$
(28)$$w_{e-z}=q_{x}\frac{\partial u}{\partial z}+q_{y}\frac{\partial v}{\partial z}+q_{z}\frac{\partial w}{\partial z}.$$

$W_{c}$, $W_{e}$, and $\left(W_{c}+W_{e}\right)$ represent the contribution of the velocity, velocity gradient and combined velocity and velocity gradient to the transport of q, respectively. These three parameters also contain three directional quota components along x, y, and z directions, and are expressed as: $W_{cx}$, $W_{cy}$, $W_{cz}$, $W_{ex}$, $W_{ey}$, $W_{ez}$ and $\left(W_{cx}+W_{ex}\right)$, $\left(W_{cy}+W_{ey}\right)$, $\left(W_{cz}+W_{ez}\right)$, respectively. The higher the value of these parameters, the greater the contribution of transporting q. $W_{cx}/∆T$, $W_{ex}/∆T$, $(W_{cx}+W_{ex})/∆T$, $q_{x}/∆T$, $W_{cy}/∆T$, $W_{ey}/∆T$, $(W_{cy}+W_{ey})/∆T$, $q_{y}/∆T$, $W_{cz}/∆T$, $W_{ez}/∆T$, $(W_{cz}+W_{ez})/∆T$, $q_{z}/∆T$ are also normalized.

Thus, the local heat-transfer coefficient was obtained as follow:(29)$$h_{local}=q_{w}/\left(T_{W}-T_{b}\right).$$

The HTP of PR = 1.5, RA = 0°, PR = 0.5, RA = 0° and PR = 0.5, RA = 10° was explained. Since MLV played an important role in heat transfer, this section mainly discusses the four parameters in the y-direction. Those values were taken from two lines formed by the intersections of the plane XOZ and planes X = x1 and X = x2 (see Figure 1d). 

As shown in Figure 8a, the value of $W_{cy}/∆T$ was close to 0 in a large region, while the value of $W_{cy}/∆T$ changed slightly in PRs and the amplitude of $W_{cy}/∆T$ increased slightly along with the X position. This means that the DWPVGs slightly changed the velocity in the y-direction behind the DWPVGs.

Figure 8b depicts the value of $W_{ey}/∆T$ in both cases and the value of $W_{ey}/∆T$ of PR = 0.5 was much stronger than PR = 1.5 near the wall and the amplitude of $W_{ey}/∆T$ became larger as it got closer to the wall along the Y direction. There was a large turning point near the wall. The value of $W_{ey}/∆T$ in X1 position was almost twice that of X2 in PR = 0.5, RA = 0°. In PR = 1.5, the trend of $W_{ey}/∆T$ was similar in both X1 and X2 positions. The contribution of the velocity gradient has a great impact on the process of transporting $q_{y}$ near the wall.

The value of $(W_{cy}+W_{ey})/∆T$ (Figure 8c) near the wall region was similar in both cases on the lines inspected, and the value of $(W_{cy}+W_{ey})/∆T$ had a large amplitude change and was less dependent on the X position. 

The variation of $q_{y}/∆T$ is shown in Figure 8d which shows that the value of $q_{y}/∆T$ in PR = 0.5 is much higher than PR = 1.5. The value of PR = 0.5, RA=0° was 40% lower than PR = 0.5,$\;RA=30^{{}^{\circ}}$ near the wall, while the amplitude of the value of $q_{y}/∆T$ did not change when the X position increased. This indicates that the longitudinal vortices played a positive role in the process of transporting $q_{y}$ only in certain areas. The figures highlight that MLV changed the intensity of $q_{y}$’s transportation.

Figure 8 showed $W_{cy}/∆T$, $W_{ey}/∆T$, $(W_{cy}+W_{ey})/∆T$ and $q_{y}/∆T$ in different cases. The MLV caused by the DWPVGs changed the contribution of the three parameters to transporting $q_{y}$ including velocity, the velocity gradient and the combined velocity and velocity gradient. MLV has different comprehensive contributions to velocity and velocity gradient and its comprehensive contribution to the transport of $q_{y}$ played a positive role locally. The most significant change took place in the local transport characteristics of $q_{y}$, meaning the existence of an MLV promoted the transport of $q_{y}$, leading to higher heat transfer performance. Thanks to the improvement of $q_{y}$’s transportation, the HTP in PR = 0.5, RA = 0° was enhanced as well, while with increased RA, the transport of $q_{y}$ was deferred. This is to say that the HTP of larger RA values was restrained.

In addition, the Nusselt number ratios ($Nu/Nu_{0}$) were introduced to evaluate the augmentation of the tube’s Nu values with the insertion of the DWPVGs, where $Nu_{0}$ was obtained from the smooth tube. Figure 9 showss that the values of $Nu/Nu_{0}$ decreased slightly with increasing Re numbers. 

### 4.2. Effect of VG’s Arrangement Styles on Friction Factor

Figure 10a shows the relationship between Re and f with different PRs and RAs. The f values were obtained according to Equation (8). The experimental results highlight that the f values were higher for the tubes with a DWPVG insertion because of the form drag from VGs. When the RA and Re were the same, the values of f increased with decreasing PR values. This is because more DWPVGs could fit into the tube if the *PR* was smaller, resulting in more form drags and severe interaction among fluid, wall, and DWPVGs. When the PR and Re were the same, the f values decreased as the RA values went down. This is because the flow blockage went up with RA values from $0^{{}^{\circ}}$ to $10^{{}^{\circ}}$ and resulted in lower speed vortices. 

Moreover, the friction efficient ratio ($f/f_{0}$) was induced to qualify the pressure drop augmentation of the tube with DWPVGs to that of the tube absence of DWPVGs (see Figure 10b). The values of $f/f_{0}$ increased with increasing Re numbers. 

### 4.3. Effects of VG’s Arrangement Styles on TEF

Figure 11 portrayed the relationship of Re and TEF with different PRs and RAs. An interesting finding was that the value of TEF decreased with the enlargement of Re. The TEF values increased with the decreasing PR values due to higher $Nu/Nu_{0}$ augmentations. The TEF values decreased with increased RA values ($0^{{}^{\circ}}\leq RA\leq 10^{{}^{\circ}}$) because of higher $f/f_{0}$ values. The maximum value of TEF was 1.21 when PR = 0.5, RA = 0° at Re=9090.

## 5. Comparison with Previous Work

Figure 12 shows the TEF in the present work compared to other works. It can be that the TEF of the DWPVGs proposed was higher than the twisted rings [40], perforated conical rings [41], conical rings [42], cross hollow twisted tapes [29], and regularly spaced quadruple twisted tapes [29]. This would mean that the DWPVGs was an effective HTE technique in the state of turbulence. The TEF of the DWPVGs was lower than that of the punched delta-winglets [43], rectangular winglets [33] and horseshoe baffles [44]. Therefore, the structure of DWPVGs needs to be further improved in future studies.

## 6. Conclusions

In this paper, new DWPVGs were proposed and studied experimentally and numerically. The effects of three pitch ratios (PR=0.5,1.0,1.5) and four rotation angles ($RA=0^{{}^{\circ}},3.3^{{}^{\circ}},6.7^{{}^{\circ}},10^{{}^{\circ}}$) on thermal-hydraulic performance were studied. The main conclusions are as follows:DWPVGs could enhance the HTP by enhancing the mixing strength of the hotter air near the wall and the colder air in the core flow region. The best HTP could be achieved by smaller PR and lower RA values.Compared with the smooth tube, the Nu and f increased by 1.24–1.83 times and 1.66–4.27 times, respectively. Considering the overall performance of the DWPVGs, the TEF was strongest when PR = 0.5, RA = 0° at Re=9090. That was TEF = 1.21.Longitudinal vortices could largely enhance the velocity gradient near the wall in the y-direction, which was the core reason for increasing the intensify of the transport of $q_{y}$.

## Figures and Tables

**Figure 1 micromachines-12-00786-f001:**
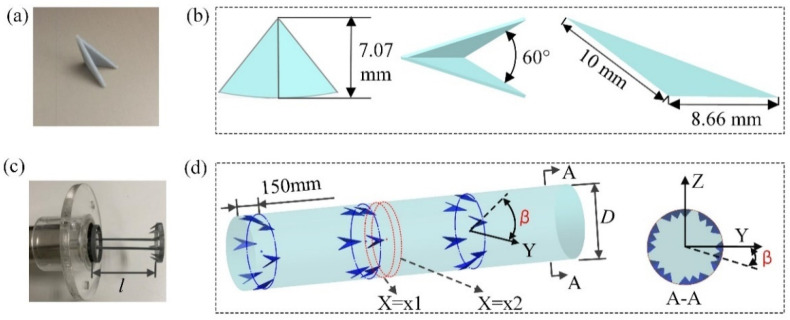
The DWPVG and test section. (**a**) Real photograph of a single DWPVG, (**b**) schematic of DWPVG, (**c**) real photograph of the test section with DWPVGs, (**d**) schematic of the test section.

**Figure 2 micromachines-12-00786-f002:**
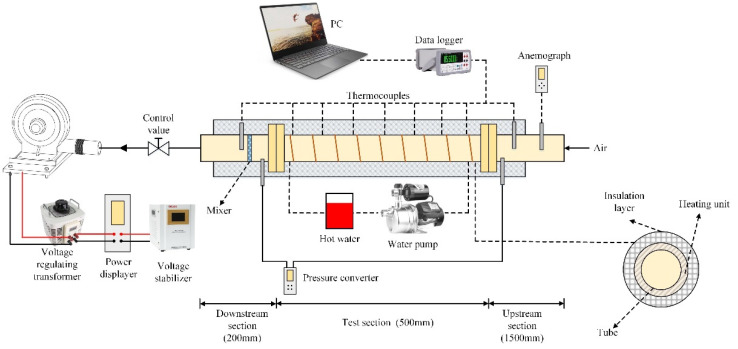
Schematic of the experiment facility.

**Figure 3 micromachines-12-00786-f003:**
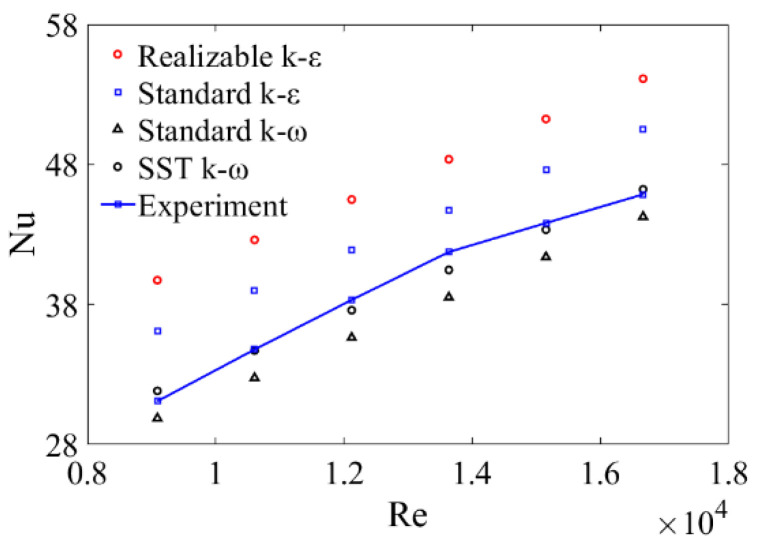
Comparison of different models of PR = 1.5, $RA=30^{{}^{\circ}}.$

**Figure 4 micromachines-12-00786-f004:**
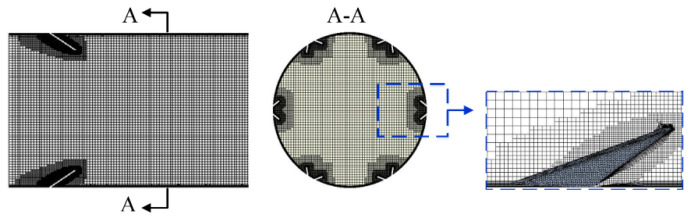
The grids generated of PR = 1.5, $RA=30^{{}^{\circ}}.$

**Figure 5 micromachines-12-00786-f005:**
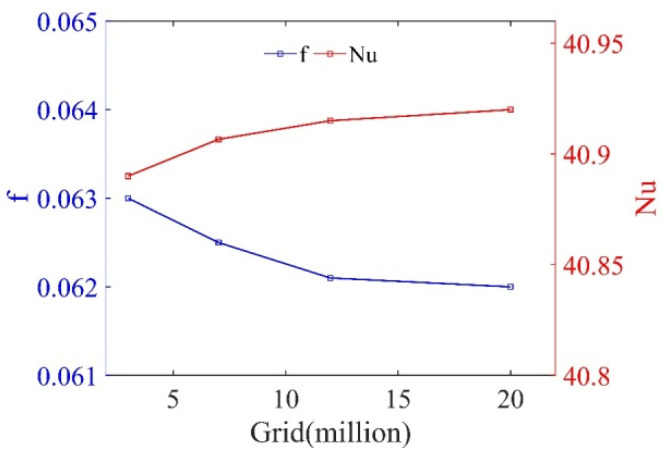
Grid independence test of PR = 1.5, $RA=30^{{}^{\circ}}.$

**Figure 6 micromachines-12-00786-f006:**
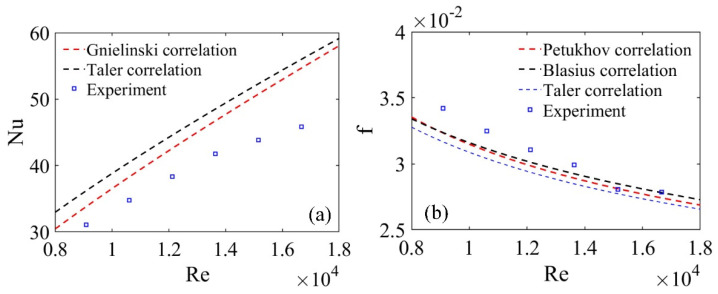
Verification of the smooth tube: (**a**) nusselt number, (**b**) friction factor.

**Figure 7 micromachines-12-00786-f007:**
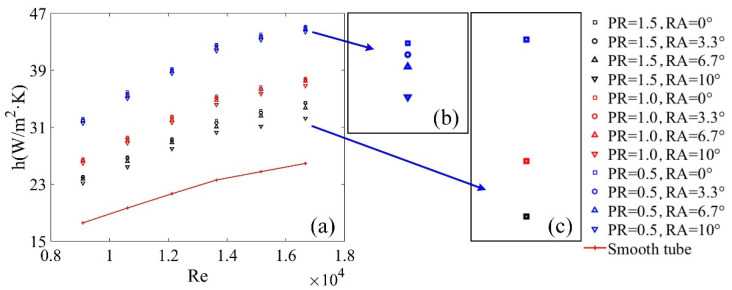
Effects of vortex generators on the *h* for different *PR*s and *RA*s. (**a**) the relationship of *Re* and h for these tubes with different *PR*s, *RA*s and *Re*s; (**b**) the relationship of *RA* and h when the values of *PR* and *Re* are the same; (**c**) the relationship of *PR* and h when the values of *RA* and *Re* are the same.

**Figure 8 micromachines-12-00786-f008:**
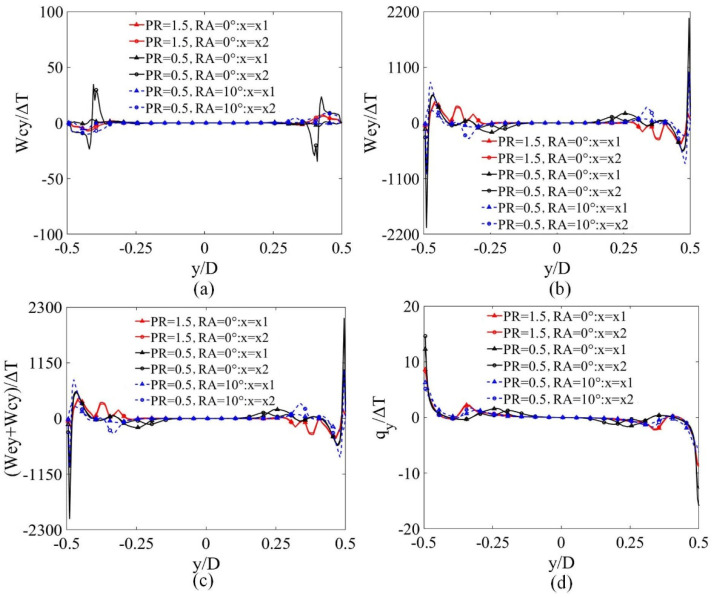
The distributions of (**a**) $W_{cy}/∆T$, (**b**) $W_{ey}/∆T$, (**c**) $(W_{cy}+W_{ey})/∆T$, (**d**) $q_{y}/∆T$ on the lines inspected.

**Figure 9 micromachines-12-00786-f009:**
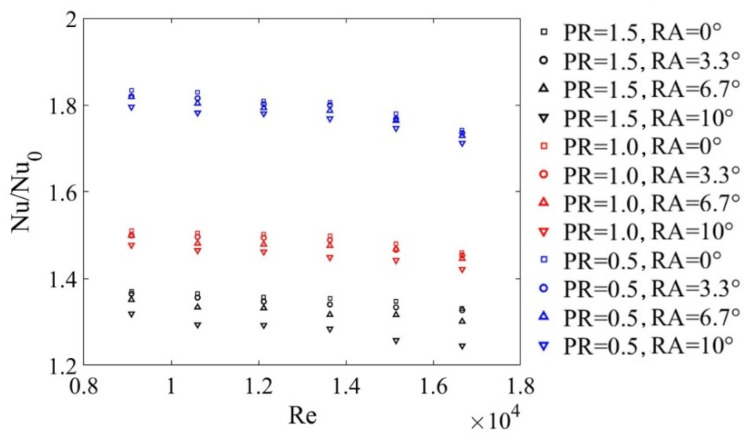
$Nu/Nu_{0}$ versus *Re* for different *PR*s and *RA*s.

**Figure 10 micromachines-12-00786-f010:**
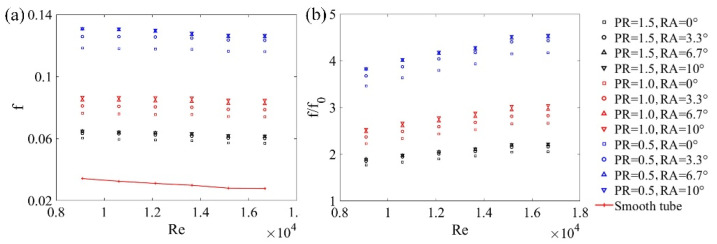
Effects of vortex generators on the *f* for different *PR*s and *RA*s. (**a**) the relationship between *Re* and *f* with different *PR*s and *RA*s; (**b**) the relationship between the friction efficient ratio $f/f_{0}$ and *Re* with different *PR*s and *RA*s.

**Figure 11 micromachines-12-00786-f011:**
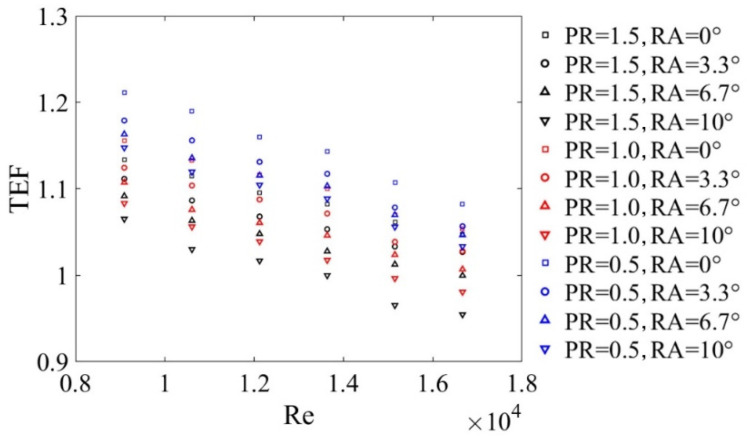
Effects of DWPVGs on the TEF for different *PR* and *RA*.

**Figure 12 micromachines-12-00786-f012:**
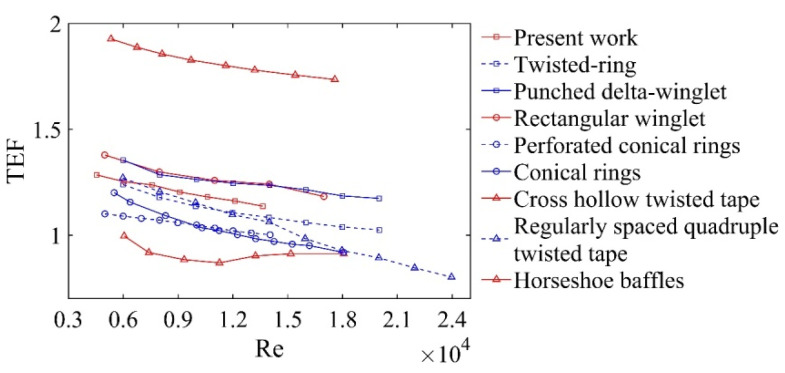
Comparisons of TEF between present work and previous works.

**Table 1 micromachines-12-00786-t001:** Boundary conditions of the simulation.

Region(s)	Conditions
Fluid inlet	$u\left(x,y,z\right)=u_{in}$ = const, $v\left(x,y,z\right)=w\left(x,y,z\right)=0,T\left(x,y,z\right)=T_{in}=293\;K$
Fluid outlet	$\frac{\partial u\left(x,y,z\right)}{\partial x}=\frac{\partial v\left(x,y,z\right)}{\partial x}=\frac{\partial w\left(x,y,z\right)}{\partial x}=\frac{\partial T\left(x,y,z\right)}{\partial x}=0,T_{out}=293\;K$
Wall surfaces	$u\left(x,y,z\right)=0,v\left(x,y,z\right)=0,w\left(x,y,z\right)=0,T\left(x,y,z\right)=T_{w}=343\;K$
DWPVGs surfaces	u(x,y,z)=0,v(x,y,z)=0,w(x,y,z)=0

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
