# Peer review of "Numerical and Experimental Investigations of Micro Thermal Performance in a Tube with Delta Winglet Pairs"

_micromachines, 2021, doi:10.3390/mi12070786_

Round 1

Reviewer 1 Report

The paper presents an interesting study and I do not  have major scientific observations/suggestions.

The main issues are related with the English language which must be carefully checked. Sentences cannot start with "And...". It is not appropriate to use "What's more.." for instance (in the Abstract).

The Introduction is a mini-review with other authors did, which is excellent, but it must be improved as language and style.

Please also avoid the repetitions (e.g. "...e energy equation, momentum equation, and turbulence equation..." could be energy, momentum, and turbulence equations.

Please keep the same style (e.g. "...?? =1.5&?? =0°, ?? =0.5&?? =0° and ?? =0.5,?? =10°..."), preferably separated by comma instead of "&".

Reviewer 2 Report

The comments are in a enclosed file.

Round 2

Reviewer 1 Report

The paper could be accepted in its present revised form.

Reviewer 2 Report

The subject of the paper are numerical and experimental investigations of micro thermal performance in a tube with delta winglet pairs.   Comments   The article focuses on the practical effects of the delta  winglet pairs. The cognitive significance of the article is low.
The article may be published in submitted form in the journal Micromachines.